# Bilateral Renal Auto-Transplantation for Retroperitoneal Sarcomas: Is It Underutilized?

Tyler P. Robinson [1], Daniel P. Milgrom [1,2], Santosh Nagaraju [1,3], William C. Goggins [1], Kannan P. Samy [1,4] and Leonidas G. Koniaris [1,5,*]

1   Department of Surgery, Indiana University School of Medicine, Indianapolis, IN 46202, USA
2   Providence Saint John's Cancer Institute, Santa Monica, CA 90404, USA
3   Renal Transplant Division, Charleston Area Medical Center, Charleston, WV 25301, USA
4   Department of Surgery, Duke University Medical Center, Durham, NC 27707, USA
5   Department of Surgery, Oregon Health and Science University, Portland, OR 97239, USA
*   Correspondence: koniaris@ohsu.edu

**Abstract:** Sarcomas are a rare tumor of mesenchymal origin. The liposarcoma is the most common sarcoma of the retroperitoneum. Liposarcomas are typically low grade, and present at an advanced stage and a large size. We report a case of a large retroperitoneal liposarcoma, approximately 50 kg, encasing both kidneys, which was managed via a two-stage resection and staged renal auto-transplantation into the intra-peritoneal pelvis. The patient maintained normal renal function throughout, and remains disease free two years post-resection. Renal auto-transplantation with pelvic placement may facilitate improved margin-free resection. Renal relocation may allow the use of curative-intent ablative therapies such as radiofrequency ablation and radiation in cases of retroperitoneal recurrence.

**Keywords:** transplantation; auto-transplantation; retroperitoneal sarcoma; sarcoma; cancer





## 1. Introduction

Soft tissue sarcomas (STSs) are a rare type of tumor of mesenchymal origin, with an incidence of 1–5 per million [1]. They are classified based on the molecular characteristics of the tumor type [2]. It is estimated that there will be over 13,000 cases of STS and 5000 deaths in 2021 [3]. Liposarcomas account for over 50% of retroperitoneal sarcomas. STS arising from a retroperitoneal location are frequently associated with a late presentation and a large size. It is not uncommon for retroperitoneal sarcomas to exceed 20–40 cm prior to the initial diagnosis. Larger retroperitoneal sarcomas are frequently low grade, as more aggressive tumors are usually diagnosed earlier, due to symptoms associated either from the local invasion or associated metastatic disease [4]. Retroperitoneal sarcomas frequently envelop or displace the kidney, meaning that their clearance is increasingly challenging, and is frequently considered a palliative undertaking. Nephrectomy may or may not be performed in this setting; however, performing nephrectomy may improve local control of the disease without progression to end-stage renal disease [5]. Ultimately, the mainstay of treatment is surgical removal. After surgical treatment, retroperitoneal sarcoma patients, overall, have a 67% 5-year survival rate [6].

## 2. Detailed Case Description

A 39-year-old male (136 kg, BMI 42) presented to an outside hospital with abdominal pain. His previous medical, surgical, and family history was notable for including diabetes, an umbilical hernia repair in 2011, and prostate cancer in a paternal grandfather. A CT of his abdomen and pelvis was obtained and demonstrated a large mass consistent with a retroperitoneal sarcoma enveloping both kidneys and displacing them anteriorly. There was also compression of the small and large bowel, and a supraumbilical hernia containing

mesenteric fat (Figure 1). This ventral hernia was reduced, and his pain resolved. The differential diagnoses for the large retroperitoneal mass included various types of sarcoma; however, this tumor displayed benign features, such as a lack of invasion into other structures. The patient was subsequently referred to a surgical oncologist at a high-volume center and saw his surgeon one week later.

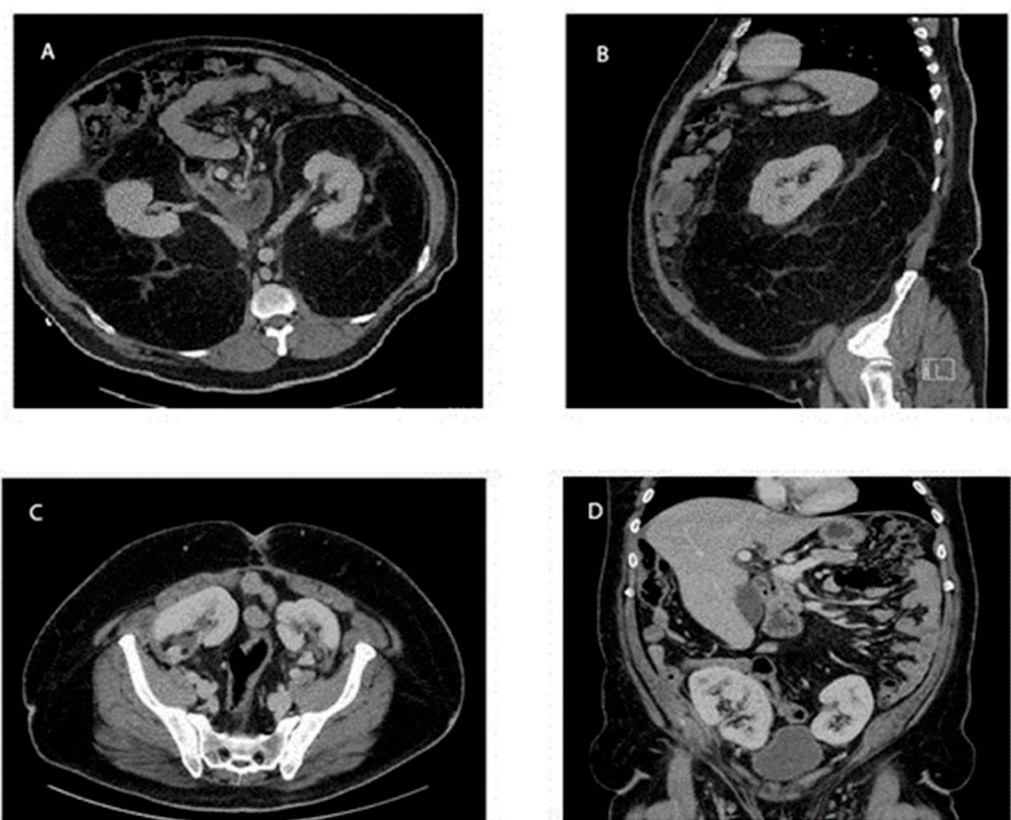

**Figure 1.** CT Scan of the abdomen and pelvis with IV contrast. (**A,B**) Pre-operative imaging demonstrating the presence of a large retroperitoneal sarcoma enveloping the bilateral kidneys. (**C,D**) Post-operative imaging at the 9 month follow up, with the bilateral kidneys now located in the pelvis, and no evidence of any recurrence of the retroperitoneal liposarcoma.

After appropriate evaluation by the surgical oncologist, risk stratification, and the preoperative optimization of his blood sugar and blood pressure, the patient was offered a two-stage operative intervention. He did not require biopsy, chemotherapy, or radiotherapy. The decision regarding whether to biopsy presented a diagnostic challenge. Due to classic imaging characteristics, expert consensus on the diagnosis, and no planned pre-operative treatment, a biopsy was deemed unnecessary, as per consensus guidelines [7]. A two-stage operation was preferred, due to the length of the procedure, and to allow the confirmation of the function of the first auto-transplanted kidney prior to the second auto-transplantation. Due to anatomical considerations, it was necessary to bisect the tumor regardless of whether the operation was performed in stages or not. Auto-transplantation allowed for the complete skeletonization of the kidney on the backbench to increase the likelihood of an R0 resection in the setting of a retroperitoneal tumor, compared to in vivo dissection, while preserving renal function and preventing tumor spillage. The patient gave informed consent and expressed a preference for this surgical approach. The first stage was the resection of the tumor on the left side, along with the left kidney en bloc, and the back-table dissection of the tumor to isolate and preserve the kidney for possible auto-transplantation. The second stage was to address the tumor on the right side in a similar fashion 4–6 weeks after recovery from the initial operation. The patient discussed

this approach with his surgical oncologist, transplant surgeon, and a urologist, who was to place renal stents for an improved identification of the ureters, as well as to facilitate the ureteral anastomosis.

The patient was admitted pre-operatively for bowel preparation with polyethylene glycol, metronidazole 500 mg, and neomycin 1000 mg. The next morning, during the first operation, a midline laparotomy incision, with a transverse "T" limb extension onto the left hemiabdomen, was used. The left colon was mobilized medially, taking care to maintain the capsule of the retroperitoneal tumor. The left renal artery and vein were identified at their respective origins from the great vessels. The ureter was identified at a point outside of the tumor. The patient was anticoagulated with 10,000 U of heparin. The left kidney and ureter were removed en bloc, with 26 kg of tumor (Figure 2). The kidney was immediately flushed with cold organ preservation (Custodial) fluid through the renal artery. The kidney was then separated from the tumor by resecting the renal capsule, as well as the fat identified in the renal hilum on the back bench. The peri-ureteric fatty tissue was resected close to the ureter (Figure 2). The left adrenal gland was retained within the sarcoma specimen and was unable to be salvaged. The left kidney was then auto-transplanted intraperitoneally into the ipsilateral pelvis, with standard renal transplant anastomoses to the left external iliac artery and vein. The ureter was anastomosed to the bladder in a mucosa-to-mucosa fashion, over a ureteral stent. The operation lasted about 9 h. The patient was started on a continuous intravenous insulin infusion to control his hyperglycemia, as well as norepinephrine and vasopressin continuous intravenous infusions to correct hypotension. On post-operative day (POD) 1, a duplex ultrasound of the auto-graft demonstrated normal-appearing kidneys, with patent vasculature and no hydronephrosis. He was transitioned to sliding-scale insulin on POD 2, and vasopressor support was discontinued on POD 3. His postoperative course was notable for continued hypotension, which was thought to be secondary to the adrenal insufficiency. This was managed with hydrocortisone 50 mg every 8 h, and midodrine 10 mg three times daily. On POD 8, the patient's morning cortisol lab value was 10.6 mCg/dL. The patient was then discharged on POD 8 with an outpatient prescription for 20 mg hydrocortisone in the morning, and 10 mg of hydrocortisone in the afternoon. He maintained normal renal function throughout his hospitalization. The pathology of the tumor was a grade 1 stage 1b T4N0 well-differentiated liposarcoma with R0/R1 margins that could not be assessed due to fragmentation in the tissue.

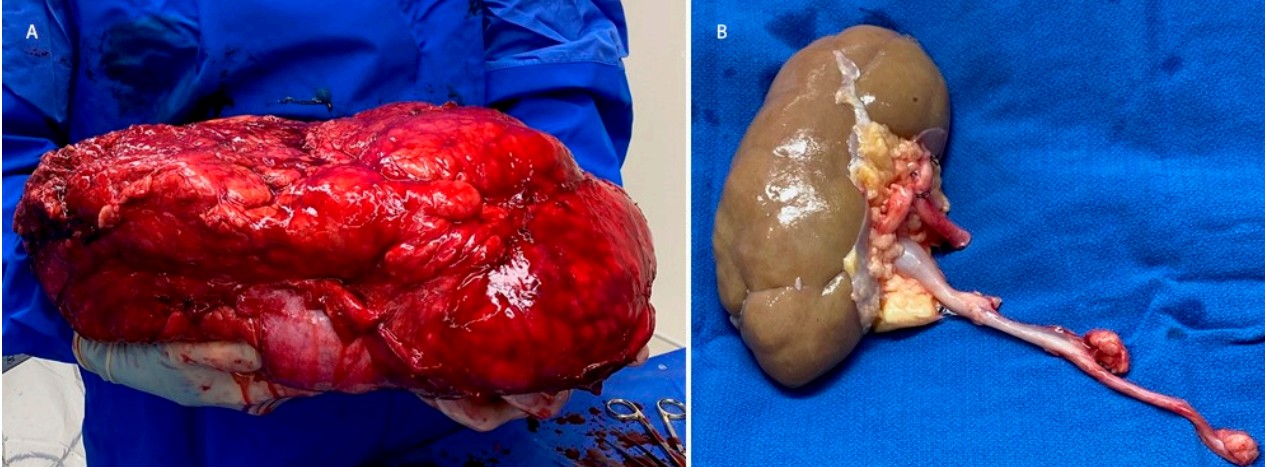

**Figure 2.** (**A**) Surgical specimen with the liposarcoma encasing the left kidney. (**B**) Left kidney after the surgical removal of the liposarcoma, prior to auto-transplantation.

On the same day as his discharge, the patient returned to the emergency department due to serosanguinous discharge from the lower aspect of his midline incision. A CT of the abdomen and pelvis was obtained, which demonstrated expected post-operative changes.

The patient was treated with a wet-to-dry wound packing on this portion of his incision. On hospital day 1, he underwent cosyntropin stimulation with values of 7.2 mCg/dL, 17.1 mCg/dL, and 19.8 mCg/dL, demonstrating adequate adrenal function. The patient was discharged on hospital day 2 with a wound vacuum over the lower aspect of his abdominal incision, and his hydrocortisone was discontinued.

Seven weeks after the initial resection, he returned to the operating room for the resection of the right retroperitoneal sarcoma with en bloc right nephrectomy and the auto-transplantation of the right kidney. He was given the same bowel preparation regimen as the first operation. The conduct of the second operation proceeded similarly to the first. The right retroperitoneal tumor was resected with the right kidney and ureter en bloc with the 23.7 kg tumor. There was a substantial displacement of the inferior vena cava, but no retrohepatic invasion of the tumor, or direct invasion into the inferior vena cava. The right kidney was prepared and auto-transplanted into the right pelvis in the same fashion as was described for the left kidney. The right adrenal gland was retained within the sarcoma specimen, leaving the patient adrenal-insufficient. This operation took about 7 h. He was immediately started on intravenous hydrocortisone 100 mg every three hours, and norepinephrine infusion for hypotension. On POD 1, his renal ultrasound again demonstrated normal-appearing kidneys, with patent vasculature and no hydronephrosis. On POD 4, his norepinephrine was discontinued. His hydrocortisone was weaned and, on POD 6, it was reduced to 25 mg twice daily and, at this point, he was started on fludrocortisone 0.1 mg daily. He was discharged on POD 14, on 15 mg hydrocortisone in the morning, and 10 mg hydrocortisone in the afternoon, as well as 0.1 mg fludrocortisone daily. Normal renal function was maintained throughout his hospitalization. Again, the pathology of the tumor was a grade 1 stage 1b T4N0 well-differentiated liposarcoma with R1/R0 margins that could not be assessed due to fragmentation in the tissue. His clinical course (Figure 3) culminated with regular clinic follow ups, and there was no evidence of any recurrence of his tumor, based upon surveillance CT scan imaging at 9 months (Figure 1), and at an outside hospital at 2 years. He continued his hydrocortisone and fludrocortisone regimen from his discharge, due to adrenal insufficiency.

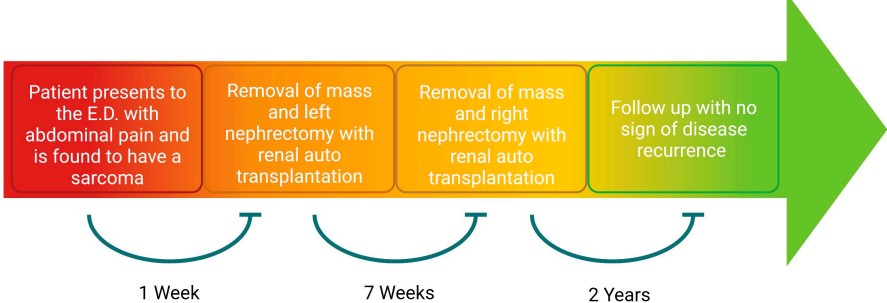

**Figure 3.** Timeline of the patient's clinical course.

## 3. Discussion

In the absence of metastatic disease, surgery remains the primary therapy for patients with STS. Retroperitoneal sarcomas carry the lowest cure and survival rates by location. This is due to their increased size, late diagnosis, and inability to obtain complete extirpation, due to their close proximity to multiple essential organs and major blood vessels [8,9]. The optimal treatment involves obtaining a margin-negative resection. Following primary resection and, frequently, with repeat resection(s), cures and significant palliation may be obtained, with survival periods of 5, 10, and more years being a realistic goal.

The management of bilateral retroperitoneal liposarcomas is extremely challenging. In the retroperitoneal location, the kidney is frequently encroached upon, or encased by, retroperitoneal sarcomas. In one study, retroperitoneal sarcomas demonstrated an absence of kidney invasion in 73% of cases, with renal capsular invasion present in 15%, parenchymal invasion in 9%, and renal vein invasion in 3% of cases [10]. The most com-

mon practice is unilateral nephrectomy. Nephrectomy leads to increased rates of acute kidney injury, acute renal failure, and a decreased glomerular filtration rate; however, end-stage renal disease and dialysis rarely occur [5,11,12]. Although many patients will do very well with a solitary kidney, organ preservation, if possible, should be the treatment objective [13,14]. Ex vivo multi-visceral explanations, tumor resections, and reimplantations have been reported anecdotally for mesenchymal tumors involving the porto-mesenteric vessels [15]. The Transatlantic Australian RPS Working Group (TARPSWG) has shown that, in the setting of a greater multi-visceral organ resection, the post-operative morbidity increases; however, this did not impact the overall survival, local recurrence, or distant metastasis of the disease. This group stressed the importance of the consideration of a multi-visceral resection, as systemic and radiotherapy efficacy remains limited [16]. Similarly, a unilateral renal auto-transplantation after the en bloc resection of a giant retroperitoneal sarcoma with a kidney has been reported (Table 1). To our knowledge, the ex vivo resection of a tumor, and the auto-transplantation of both kidneys, has not been reported for mesenchymal stromal tumors. The ex vivo resection and reimplantation of the kidney has three theoretical benefits. It provides an improved ability to clear the tumor around the kidney, potentially with less blood loss and tumor spillage. Kidney auto-transplantation moves the kidney from the retroperitoneal location, preventing recurrences from impacting the kidney and, potentially, making subsequent resections easier if, or when, needed. Thirdly, moving the kidney to the intraabdominal pelvis would allow the easier and potentially safer application of ablative therapies, such as radiation and radiofrequency ablation, to the region of the retroperitoneum, should recurrence occur.

**Table 1.** Case reports describing sarcoma resections and renal auto-transplantations.

| Author | Year | Case Description | Follow Up | Disease Status |
|---|---|---|---|---|
| Paloyo et al. | 2019 | A 47 × 34 × 17 cm 11 kg tumor encased the right kidney and ureter, requiring a right nephrectomy and auto-transplantation. The pathology demonstrated a low-grade, well-differentiated liposarcoma. Nuclear imaging demonstrated good renal function post-operatively [17]. | 6 months | No recurrence |
| Fernandez et al. | 2015 | A 13.5 cm tumor, encasing zone I–III of the IVC, requiring the resection of the IVC, a complete hepatectomy, and a bilateral nephrectomy with IVC reconstruction via a vascular graft, and the auto-transplantation of the liver and the left kidney. The pathology demonstrated a high-grade spindle cell sarcoma of vena cava origin. The liver function was found to be within normal limits, and the baseline creatinine increased to 1.8 [18]. | 1 year | No recurrence |
| Bansal et al. | 2013 | A 40 × 35 × 35 cm 24 kg tumor, involving the right ureter and ileum, ultimately requiring a small bowel resection and a right nephrectomy with auto-transplantation. The pathology demonstrated a mixed-type liposarcoma. Post-operatively, nuclear imaging demonstrated good renal function [19]. | 63 months | Recurrence at 40 months, requiring reoperation. No further recurrence as of the 63 month follow-up |
| Kraybill et al. | 1997 | Two patients were described as having a sarcoma of the IVC, requiring the resection of the IVC and aorta, and a bilateral nephrectomy, via the vascular graft reconstruction of the IVC and aorta, and left renal auto-transplantation. The first patient had a 15 × 10 × 7 cm poorly differentiated spindle cell rhabdomyosarcoma. Post-operatively, she showed recovery in her creatinine to near the baseline. The second patient had a 4.5 cm moderately differentiated leiomyosarcoma. Post-operatively, her creatinine increased to 1.5 [20]. | Patient 1—8 months Patient 2—about 2.5 years | Patient 1—Recurrence in the right psoas muscle; patient declined further treatment Patient 2—Recurrence in the lung and liver |

The potential benefit of chemotherapy for STSs likely depends on the tumor type [21]. A small benefit from chemotherapy is generally limited to higher-grade tumors. Adjuvant chemotherapy for high-risk RPSs is currently being evaluated, in the STRASS2 trial. There is no or minimal benefit from adjuvant radiation to patients with retroperitoneal STS, as demonstrated by the STRASS trial [22]. TARPSWG suggests considering neoadjuvant radiation in tumors with a high risk of local recurrence, such as well-differentiated liposarcomas

and low-grade dedifferentiated liposarcomas [7]. Likewise, TARPSWG recommends the consideration of chemoradiation with cytoreductive intent in highly selective cases [7]. We posit that, depending upon the potential recurrence sites, patients who have undergone kidney relocation to the pelvis might be candidates for radiation, as well as newer therapies, such as radiofrequency ablation. The use of such adjuvant technologies may be facilitated by the absence of the kidney from the retroperitoneal space.

Due to the clear benefit provided by surgery, and the marginal benefit provided by adjuvant therapies, approaches that optimize an R0 surgical resection for retroperitoneal sarcoma patients should be prioritized. One potential obstacle to the implementation of newer surgical technologies for STS is that many of the current treatment strategies have come from free-standing cancer centers that, in many instances, are without access to non-oncologic surgical disciplines and technologies, such as organ transplantation and advanced vascular surgery [23]. This technique is also limited by careful patient selection. We present the first reported case of a giant retroperitoneal liposarcoma encasing both kidneys managed via the resection and auto-transplantation of both kidneys. The patient showed no evidence of recurrence at his 2-year follow-up.

### 4. Conclusions

Patients with large retroperitoneal sarcomas are best treated with surgery. When the tumor encases the kidneys, the removal of the tumor via renal auto-transplantation is an option to attempt an R0 resection.

**Author Contributions:** Conceptualization, L.G.K.; writing—original draft preparation, T.P.R.; writing—review and editing, D.P.M., S.N., L.G.K., K.P.S. and W.C.G.; visualization, T.P.R. and S.N.; supervision, L.G.K.; funding acquisition, L.G.K. All authors have read and agreed to the published version of the manuscript.

**Funding:** Funded by the National Institute of General Medical Sciences, grant number 5R01GM137656-02 and the National Cancer Institute, grant number 5P01CA236778-02 5266.

**Institutional Review Board Statement:** Not applicable.

**Informed Consent Statement:** Written informed consent has been obtained from the patient to publish this paper.

**Data Availability Statement:** Not applicable.

**Acknowledgments:** Figure 3 was created using BioRender.com.

**Conflicts of Interest:** The authors declare no conflict of interest.

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
