# Peer review of "Bilateral Renal Auto-Transplantation for Retroperitoneal Sarcomas: Is It Underutilized?"

_curroncol, doi:10.3390/curroncol30080552_

Round 1
Reviewer 1 Report
The article by Tyler and Coll is a classical case report, difficult to be exported or reproduced, but gives some suggestions in the treatment of retroperitoneal soft tissue sarcomas. (RPS)
The introduction is fine and well circumscribes the situation: a huge bilateral liposarcoma of the RP enveloping both kidneys is the case.
The decision to make a renal autotransplant was due to the complete involvement of the kidneys .
The case description is complete and the difficult postoperative situations with the following complications are well reported. The first conclusion is that such a solution can be purposed only in very limited and particular cases and cannot be considered as an abitual surgical approach.
The transplant should be done in a Referral center for sarcomas.
In the discussion a wider description of the multivisceral intervention as the Transatlantic Group stated should be taken in consideration.
Always in the discussion the postoperative radiotherapy is reported as a possible adjuvant approach. This is excluded from the NCCN and ESMO guideliness , mainly in low grade liposarcoma because of its inefficacy. Nowadays radiotherapy can be considered only in a neadjuvant setting , in selected cases. ( STRASS study The Lancet Sept 2020).
This and similar publications can be cited in the bibliography.
Author Response
Current Oncology Case Report Response
8/7/23
Response to Editor:
We would like to thank the editor for their comments. In response to comment 1 we cite The Transatlantic Australian RPS Working Group(TARPSWG) guidelines surrounding indications for a biopsy. We have changed the statements noted by the reviewer to now read the following on lines 55-56:
“Due to classic imaging characteristics, expert consensus on the diagnosis, and no planned preoperative treatment, a biopsy was deemed unnecessary per consensus guidelines[7].”
In response to comment 2 during our operation, we were fortunate to be able to avoid multi-visceral resection as the tumor did not involve any abdominal organs other than the bilateral kidneys and adrenal glands. Instead, the tumor displaced organs and structures such as the mesentery. We have added increased discussion of multi-visceral resection on lines 165-170:
“The Transatlantic Australian RPS Working Group(TARPSWG) has shown that in the setting of a greater multi-visceral organ resection, post operative morbidity increases, however this did not impact overall survival, local recurrence, or distant metastasis of the disease. This group stressed the importance of consideration of a multi-visceral resection as systemic and radiotherapy efficacy remains limited [16].”
In response to comment 3, we have added discussion of the Transatlantic Group recommendations regarding radiotherapy, which cite the STRASS trial on lines 185-187:
“TARPSWG suggests to consider neoadjuvant radiation in tumors with high risk of local recurrence such as well-differentiated LPS and low grade dedifferentiated LPS[23].”
Again, we would like to thank the editor and feel that the editor’s suggestions have improved the quality of the manuscript.
Response to Reviewer 1:
We appreciate the reviewers comments regarding the content and discussion within our manuscript. Further, we have attempted to incorporate suggestions by the reviewer for the manuscript. We have cited the Transatlantic Group and provided a more thorough description on lines 165-170 as follows:
“The Transatlantic RPS Working Group has reported that in the setting of a greater multi-visceral organ resection, post operative morbidity increases, however this did not impact overall survival, local recurrence, or distant metastasis of the disease. This group stressed the importance of consideration of a multi-visceral resection as systemic and radiotherapy efficacy remains limited [16].”
We have added further citations to The STRASS study (line 184-185), mentioned the STRASS2 study and the information it will provide on chemotherapy recommendations in high-risk RPS(line 183-184.) Finally, we have added discussion of The Transatlantic Australian RPS Working Group(TARPSWG) and their recommendations regarding chemotherapy and radiation.
Thank you for your comments regarding our manuscript.
Response to Reviewer 2:
We also appreciate this reviewers comments which summarize our manuscript and the areas for improvement which the reviewer identifies. We have included a more robust discussion of improved local recurrence and renal injury vs failure in the introduction and discussion on lines 34-36 which states:
“Nephrectomy may or may not be performed in this setting, however performing nephrectomy may improve local control of the disease without progression to end-stage renal disease[5].”
As well as lines 159-161 which state:
“Nephrectomy leads to increased rates of acute kidney injury, acute renal failure, and decreased glomular filtration rate, however end stage-renal disease and dialysis do not occur[5,11,12].”
We have attempted to strengthen the paragraph that discusses chemotherapy and radiation by adding the following to lines 183-189:
“Adjuvant chemotherapy for high-risk RPS is currently being evaluated in the STRASS2 trial. There is no or minimal benefit for adjuvant radiation for patients with retroperitoneal STS, as demonstrated by the STRASS trial[22]. TARPSWG suggests to consider neoadjuvant radiation in tumors with high risk of local recurrence such as well-differentiated LPS and low grade dedifferentiated LPS[23]. Likewise, TARPSWG recommends consideration of chemoradiation with cytoreductive intent in highly selective cases[23]. “
Thank you very much for reviewing our manuscript and providing your comments.
Reviewer 2 Report
The authors present a case study of a 39 year old male with diabetes who was suspected to have a retroperitoneal sarcoma (RPS) after presenting to an emergency department with abdominal pain. The patient was identified to have a noninvasive retroperitoneal mass of undifferentiated origin displacing bilateral kidneys and compressing both large and small bowel. The authors make an argument suggesting that this case is worth reporting due to their somewhat novel solution to a frequently encountered challenge created by RPS: 1. Dangerous dissection planes with risk of damaging the kidney and its associated structures when attempting to spare the organ and 2. How to preserve renal function in the setting of a retroperitoneal mass which would otherwise require bilateral nephrectomy and lifelong dialysis. They go on to describe a two-staged approach in which the left half of the mass and the left kidney is removed en bloc, skeletonized, and auto-transplanted with the contralateral side addressed in identical fashion once the patient has recovered from the index operation. The tables and figures are extremely effective, and the literature review provided in Table 1 is thorough.
Overall, I have no major concerns with the contents of this manuscript, and certainly the approach has only been described a handful of times in the literature (generally for unilateral renal autotransplantation). My comments that may assist the authors in improving the work are as follows:
The overarching argument of the manuscript is that this approach will preserve renal function and reduce recurrence. Would recommend that relevant data including expected rates of renal injury vs failure (with subsequent need for dialysis) for RPS resections be discussed in the Introduction and Discussion to more thoroughly .
The discussion in the paragraph beginning on Line 170 is unclear and requires reframing. It appears the message is that adjuvant therapy for RPS is generally ineffective, and that since surgical resection remains the mainstay of curative intent treatment novel approaches to optimize R0/R1 resection at the initial presentation should be prioritized. Perhaps this paragraph could be combined with the following one to build that argument and then continue with justification of why a focus on surgical technique matters. Citation of the recent STRASS trial (reviewer not an author on that study) is also recommended to justify the statement that radiation may have a limited benefit in RPS since for years it was heavily utilized in the neoadjuvant space prior to this data.
Author Response

(The authors gave the same response as above.)

Round 2
Reviewer 1 Report
Congratulations to the Authors for the revision.
The update of the references is well done.
No further remarks
Reviewer 2 Report
Revisions are well received. No further comments for improvement of the article.